# Threshold Interphase Delay for Bipolar Pulses to Prevent Cancellation Phenomenon during Electrochemotherapy

**DOI:** 10.3390/ijms25168774

**Published:** 2024-08-12

**Authors:** Veronika Malyško-Ptašinskė, Aušra Nemeikaitė-Čėnienė, Eivina Radzevičiūtė-Valčiukė, Eglė Mickevičiūtė, Paulina Malakauskaitė, Barbora Lekešytė, Vitalij Novickij

**Affiliations:** 1Faculty of Electronics, Vilnius Gediminas Technical University, 10223 Vilnius, Lithuania; eivina.radzeviciute@imcentras.lt (E.R.-V.); egle.mickeviciute@imcentras.lt (E.M.); paulina.malakauskaite@imcentras.lt (P.M.); barbora.lekesyte@imcentras.lt (B.L.); 2Department of Immunology and Bioelectrochemistry, State Research Institute Centre of Innovative Medicine, 08406 Vilnius, Lithuania; ausra.ceniene@imcentras.lt

**Keywords:** cancellation effect, interphase delay, PEF, cisplatin, electrochemotherapy, permeabilization, in vitro, viability

## Abstract

Electroporation-based procedures employing nanosecond bipolar pulses are commonly linked to an undesirable phenomenon known as the cancelation effect. The cancellation effect arises when the second pulse partially or completely neutralizes the effects of the first pulse, simultaneously diminishing cells’ plasma membrane permeabilization and the overall efficiency of the procedure. Introducing a temporal gap between the positive and negative phases of the bipolar pulses during electroporation procedures may help to overcome the cancellation phenomenon; however, the exact thresholds are not yet known. Therefore, in this work, we have tested the influence of different interphase delay values (from 0 ms to 95 ms) using symmetric bipolar nanoseconds (300 and 500 ns) on cell permeabilization using 10 Hz, 100 Hz, and 1 kHz protocols. As a model mouse hepatoma, the MH-22a cell line was employed. Additionally, we conducted in vitro electrochemotherapy with cisplatin, employing reduced interphase delay values (0 ms and 0.1 ms) at 10 Hz. Cell plasma membrane permeabilization and viability dependence on a variety of bipolar pulsed electric field protocols were characterized. It was shown that it is possible to minimize bipolar cancellation, enabling treatment efficiency comparable to monophasic pulses with identical parameters. At the same time, it was highlighted that bipolar cancellation has a significant influence on permeabilization, while the effects on the outcome of electrochemotherapy are minimal.

## 1. Introduction

Electrochemotherapy (ECT) has emerged as a promising modality in cancer therapy, combining the cytotoxic effects of chemotherapeutic agents with the cell-permeabilizing properties of pulsed electric fields (PEFs) to treat various cancer types [1,2,3] and their anatomical sites [4,5,6,7]. The efficiency of ECT directly depends on both cytotoxic anti-cancer drugs and the properties of PEF, such as waveform, duration, pulse number, or burst delivery frequency. In ECT clinical trials, eight 0.8–1.4 kV/cm × 100 µs duration pulses delivered at 1 Hz repetition frequency remain the predominant pulsing protocol [8,9,10], known as ESOPE. Nonetheless, negative aspects associated with such pulsing protocols have been identified, including non-homogeneous treatment [11], the potential for tissue damage (due to thermal or oxidative damage) [12], and mild discomfort or muscle contractions [13]. Overall, while the ESOPE pulsing protocol has demonstrated good efficacy, it is important to search for alternative treatment approaches and improve the methodology even further.

Studies suggest that it is possible to minimize the mentioned negative effects and, at the same time, enhance the efficacy of the treatment itself by manipulating various PEF parameters [14]. This includes reducing the pulse duration (to sub-microseconds) [15], increasing the pulse repetition frequency [11,16,17], or/and switching the phase polarities [18] to increase effect homogeneity due to the higher frequency component of the burst and impedance mitigation. The last-mentioned approach involves the utilization of electroporators capable of generating high-frequency nanosecond bipolar pulses [19]. Bipolar pulses comprise two phases with opposite polarities. This configuration potentially decreases the net charge delivered to the tissue compared to unipolar pulses [20], which consist of a single polarity [21], thus reducing the possibility of tissue damage and minimizing the induction of electrolysis and pH changes [22]. Additionally, significant minimization of muscle contractions is ensured [23], and impedance mitigation [24] attributes to a more uniform spatial electric field distribution and a more homogeneous treatment, which makes the utilization of bipolar pulses promising. As a result, bipolar electroporation may become a valuable technique in medical and biological applications, including drug delivery, gene therapy, and cancer treatment. Therefore, the primary motivation for employing bipolar PEFs in ECT procedures is the advantages they offer, which may lead to more efficient treatment compared to the currently used PEF protocols in clinical settings.

However, when nanosecond pulses are delivered in symmetrical sequences, they are less effective when compared to an identical unipolar burst [25]. This phenomenon, known as the bipolar cancellation effect (BPC), occurs when the second opposite polarity pulse in the burst cancels out the effects of the previous one. Studies suggest that there are various effects included in the development of BPC. One hypothesis, supported by theoretical analysis [26], suggests the reverse electrophoretic transport of Ca^2+^ ions across the plasma membrane and out of the cell. However, Gianulis et al. have demonstrated that reversal of electrophoretic flows of Ca^2+^ ions may not be the issue [27]. This phenomenon might also be linked to an increased membrane discharge time. This is because the opposite pulse accelerates the discharge, thereby diminishing the effect of nanosecond pulsed electric fields and decreasing cell molecular uptake [28]. Therefore, variation in pulse shape may play a pivotal role in cell membrane permeability using bipolar pulses.

While most studies on this topic mainly focus on irreversible electroporation [29,30], data on bipolar electrochemotherapy are scarce. Based on current knowledge, it is presumed that modifications to the bipolar sequences are necessary to overcome BPC and enable effective ECT [31,32]. One of the solutions could be introducing pulse asymmetry, i.e., different amplitude values or durations of opposite phases [33]. Valdez et al. demonstrated that complete elimination of BPC can be achieved by using a shorter duration for the first positive pulse compared to the subsequent [33] negative pulse, with durations of ↑300 ns and ↓900 ns, respectively. However, employing the opposite pulse sequence of ↑900 ns followed by ↓300 ns resulted in the persistence of cancellation. Another approach is incorporating an interphase delay between opposite polarity phases. Researchers concluded that a 10 ms interphase delay can elicit a cell permeabilization rate comparable to one triggered by a unipolar pulsing protocol [34]. Similar findings were reported by Pakhomov et al.; however, they observed mitigation of BPC with as little as a 10 µs [25] or 50 µs [28] delay in their studies, respectively. However, despite the benefits offered by the introduced methods for minimizing BPC, a comprehensive explanation has not yet been provided.

On the other hand, ECT success also strongly depends on chemotherapeutic drugs. Currently, bleomycin (BLM) [35,36,37] and cisplatin (CDDP) [38,39,40] are dominating the field; however, their transport into cells when combined with nanosecond pulsed electric fields (nsPEF) differs. This is attributed to the relationship between pore size, its density during PEF exposure, and the size of the drug molecule. Nanosecond bursts generate smaller pores compared to those created by microsecond pulses (such as ESOPE) [41,42]. However, depending on burst properties, the number of nanopores may be greater [43], allowing for the uptake of small molecules or ions more efficiently via the concentration gradient [44]. The death of a cell occurs when it contains enough anti-cancer agent molecules to enter the cell interior [45]. Simultaneously, it is also important to ensure a minimal concentration that does not affect cell viability in the absence of an electric field [46]. When comparing CDDP and BLM, it is worth noting that BLM is considered to be a larger molecule than CDDP, featuring 100 nm and 1 nm, respectively [43,47,48,49]. Bleomycin is regarded as more efficient when utilized with microsecond PEFs; however, Vižintin et al. demonstrated that a higher concentration of bleomycin was necessary to achieve a significant reduction in cell viability when used with nanosecond bursts [50]. Additionally, Scuderi et al. determined that the quantity of cisplatin molecules needed for effective ECT with ESOPE is comparable to that required for nanosecond pulses [51]. As a consequence, CDDP may be considered an appropriate anticancer agent when combined with nsPEF.

In this study, for the first time, we characterized the effect of interphase durations ranging from 0 to 95 ms with a 10 Hz pulse repetition frequency and from 0 to 0.5 ms with 100 Hz and 1 kHz bursts, respectively, on bipolar pulses of 300 ns and 500 ns duration. Our results showed that there is a threshold delay value that allows mitigation of the bipolar cancellation effect in all the tested frequency ranges, enabling molecular uptake equivalent to that obtained with unipolar pulses. Subsequently, CDDP-based ECT efficiency employing symmetrical bipolar pulses with 0 and 0.1 ms interphase delay and unipolar pulses, all delivered at a 10 Hz pulse repetition frequency, was compared to ESOPE procedures.

## 2. Results

### 2.1. Cell Membrane Permeabilization

To assess the electroporation efficiency, the permeabilization of the cell membrane with unipolar pulses was characterized first. We have analyzed the permeabilization rate while applying 300 and 500 ns duration pulses of PEF strength from 0 to 13 kV/cm with a 10 Hz repetition frequency. The results are shown in Figure 1. It can be seen that the scaling with PEF amplitude is present. As expected, increasing the pulse duration also increases the permeabilization rate.

According to Figure 1, high permeabilization (~75%) can be achieved with a pulse amplitude above 7 kV/cm and a pulse duration of 500 ns, when delivered in 50 pulse bursts (10 Hz). Based on the permeabilization results, next we assessed the uptake of YP resulting from the delivery of bipolar pulses (↑300 ns + Δt + ↓300 ns and ↑500 ns + Δt + ↓500 ns) while gradually increasing the interphase delay Δt from 0 up to 95 ms, using the same electric field strength values (4, 7, 10, and 13 kV/cm) (Figure 2).

As anticipated, the application of 4 kV/cm did not elicit any statistically significant cell response to PEF, regardless of whether a pulse duration of 300 ns or 500 ns was utilized. The observed outcomes were comparable to those of the untreated control samples. Regarding the remaining samples, the uptake of YP increased in a dose-dependent manner with PEF, where higher amplitudes (7–13 kV/cm) and longer durations of PEF resulted in greater cell permeabilization.

Although a statistically significant difference in YP uptake was observed between samples exposed to PEFs with a 0 ms interphase delay and those with delays ranging from 0.1 to 95 ms, it was shown that delivering the pulses without delay triggers a cancellation phenomenon, leading to a reduction in cell permeabilization of approximately 15–30% YP uptake (considering the standard deviation) across all tested protocols. No notable difference in cell membrane permeabilization efficacy was noted when interphase delays above 0.1 ms were used.

Further, the effects of frequency on cell membrane permeabilization were characterized to determine if the pulse repetition rate (10 Hz–1 kHz) affects the cancellation phenomenon. The unipolar pulses were used, and the results are summarized in Figure 3.

Based on the results, it is concluded that pulse repetition frequency has no significant effect on cell membrane permeabilization in the described range of parameters. The efficacy of 100 Hz and 1 kHz protocols is equivalent to 10 Hz procedures, while cell membrane permeabilization depends on PEF amplitude and duration. The cells were also subjected to PEF using the 1.2 kV/cm × 100 μs × 8 pulsing protocol (ESOPE), resulting in 90 ± 5% of YP fluorescent cells (depicted by the red dashed line in Figure 1) being further used in the study as a positive reference for electrochemotherapy.

Lastly, we assessed the effects of frequency on the permeabilization triggered by bipolar sequences (Figure 4).

The study was limited to a 7 and 10 kV/cm electric field to prevent saturation in response. The interphase delays of 0, 0.1, and 0.5 ms were tested. Based on the results obtained, the most significant cancellation effect occurs when zero interphase delay is used, regardless of pulse duration, amplitude, or exposure time. Across all protocols, employing a 0 ms delay between the positive and negative phases reduces YP uptake by 15–45% (taking into account the standard deviation). Similar to the unipolar pulse case, the effects of pulse repetition frequency are low in the investigated parametric range. It can be seen from Figure 4 that differences in permeabilization efficacy between 10, 100 Hz, and 1 kHz pulses are usually in the sub-15% range. However, it is confirmed that there is a threshold interphase delay of 0.1 ms when the efficacy of bipolar pulses is comparable to unipolar bursts (Figure 3).

### 2.2. Electrochemotherapy with Cisplatin

The effects of 7 and 10 kV/cm protocols were assessed in the context of cisplatin-based electrochemotherapy. We have evaluated the effect of 7 and 10 kV/cm pulsing protocols on cell viability without cisplatin (Figure 5, gray columns). The 7 kV/cm protocols resulted in predominantly reversible electroporation (viability drop of 10–30%); however, the 10 kV/cm pulses triggered higher electroporation with a viability drop higher than 30% (for 500 ns pulses, Figure 5B).

As expected, the application of PEF with cisplatin can several-fold potentiate the cytotoxic efficacy. It is evident that the majority of applied protocols yielded efficiency levels similar to those of ESOPE. In most of the cases, the results revealed no significant differences in cell viability between samples treated with unipolar and bipolar bursts. Also, the effect of delay (0 or 0.1 ms) did not affect the efficacy of ECT or it was lost within the standard deviation of data. Although these protocols featured 15–30% variation in YP uptake (Figure 4), the effects on cell viability and ECT were negligible (Figure 5). However, it should be noted that modulation of frequency might impact the ECT efficiency to some extent [52]. 

## 3. Discussion

This study is focused on investigating the role of different interphase delay values on BPC and the efficiency of cisplatin-based ECT in vitro using the mouse hepatoma MH-22a cell line as a model. We have shown that a cancellation effect is present when the interphase delay is low; however, once the delay exceeds 0.1 ms, the efficacy of bipolar pulses becomes comparable with unipolar ones. No statistically significant difference in permeabilization rate was noticed by increasing the delay up to 95 ms. Valdez et al. determined that 10 ms are required to mitigate BPC and ensure a high level of YP-1 uptake in CHO-K1 cells [34]. Delays that are shorter than 10 ms result in a 2-fold or lower permeabilization, which is not the case in our study. This could be attributed to the different cell lines, buffers, and pulses employed in our work. Indeed, the BPC phenomenon also depends on the duration of the pulse; e.g., a 10 µs delay between opposite polarity pulses is sufficient to achieve the effect comparable to single polarity PEFs if ↑600 ns + delay + ↓10 µs [25,26] bursts are used. Similarly, Valdez et al. [33] have shown that a longer duration of the negative phase proved beneficial to ECT compared to symmetrical bipolar pulses with no interphase delay. Nevertheless, the most effective were asymmetrical PEFs where interphase was introduced, indicating that variability in pulse sequence form is advantageous for minimizing BPC.

We have also shown that the effects of pulse repetition frequency (10 Hz, 100 Hz, and 1 kHz) were minor on permeabilization rate of the cells within the studied parameters. Several studies have analyzed the ECT efficacy while applying >1 Hz pulse repetition frequency. Certainly, the effectiveness of ECT fluctuates with changing pulse repetition frequencies [53]; yet, when increasing the pulse repetition frequencies to several kilohertz, the uptake of external molecules remained at a similar level as with the standard frequency of 1 Hz [54,55], which is consistent with our findings. Our results indicate that frequencies up to 1 kHz do not affect the BPC phenomenon. However, due to the potential for more homogeneous treatment [53] and fewer adverse effects on patients [56], the field is shifting towards the application of higher-frequency PEFs. Biphasic pulses of the same frequency, when compared to monophasic bursts, feature a higher frequency component, potentially allowing more efficient ECT due to impedance mitigation. Rembiałkowska et al. have evaluated the BPC possibility when compressing pulses to MHz [57]. Their research revealed that although MHz bursts are very efficient when utilizing monophasic pulses, bipolar sequences of the same frequency result in complete cancellation (not the case in our study). It is indicative that the phenomenon and the extent of cancellation are limited in time (it is a dynamic process)—the higher the bipolar burst frequency, the more profound the cancellation phenomenon.

Polajžer et al. [58] demonstrated that when utilizing short pulses (supra-electroporation), the cell membrane does not fully charge before the application of the opposite polarity pulse. This results in reduced molecular uptake and higher survival, but only with a short interphase delay, which may be attributed to incomplete discharge. Conversely, a longer delay of 100 µs is sufficient; as a consequence, longer pulses (0.5–5 µs) and longer interphase delays enable a full charge/discharge cycle and thus result in a more efficient procedure. Our results are in agreement with the hypothesis.

In the context of ECT with cisplatin and bipolar pulsed electric fields, our results demonstrate that there is no significant difference in cell viability between samples treated with pulse bursts utilizing monopolar or bipolar bursts (within the investigated parametric range). It is indicative that the cancellation phenomenon affecting 20–30% permeabilization of YP is not game-changing in terms of drug delivery. It might be more profound in the case of drugs of higher molecular size, but nano-range ECT with cisplatin results in similar efficacy as ESOPE, which is in agreement with the study by Vižintin et al. [59]. In cases of higher-frequency bursts (1 MHz or higher), the cancellation is more dramatic and should be compensated [57].

It should be noted that bipolar PEF protocols result in a higher input energy of the burst when compared to monophasic bursts (i.e., double); however, the cells respond similarly, as no difference in cell survival was detected between bipolar and unipolar PEFs. However, potentially, it might introduce challenges in terms of the management of Joule heating. De Caro et al. summarized that while the compression of electric pulses leads to a decrease in pain and electrochemical effects, the thermal effect is amplified [30]. Another important consideration is that the interphase delay not only affects electroporation efficacy but also influences muscle contractions, pain, and nerve stimulation. A study by Cvetkoska et al. demonstrated that short high-frequency biphasic pulses with a short interphase and long inter-pulse delays can mitigate both muscle contractions and pain sensations compared to pulses delivered with a long interphase and short inter-pulse delays [60]. Therefore, despite the potential for enhanced ECT efficiency with longer interphase delays, it is advisable to select the shortest possible value to minimize adverse effects.

In clinical practice, ECT is combined with other cancer treatment methods, such as radiotherapy (RT), to ensure profound action against cancer. Interestingly, electroporation alone exhibits a radiosensitizing effect, allowing for a specific period (lasting 10 to 50 min [61]) of enhanced radiosensitivity, enabling the application of a safe, reduced dose of irradiation. Subsequently, this process may enhance the efficiency of radiotherapy and improve tumor eradication by increasing toxicity [62]. Studies suggest that the combination of ECT with irradiation may contribute to the increased radiosensitizing effect of anti-cancer drugs such as BLM [63], CDDP [64], and other agents [65] using the ESOPE established PEF protocol and, thus, reduced tumor viability. The first clinical trial investigating the combination of bleomycin-based ECT with RT was published by Skarlatos et al. [66]. The treatment demonstrated a complete response in 63.83% of various tumors, a partial response in 31.91%, and an overall response rate of 95.74% for the treated nodules. Patients experienced neither systemic nor local side effects.

Though the effect of nsPEFs on radiosensitization has not yet been studied, the combination of sub-microsecond bipolar pulses and radiotherapy has only been studied in the context of high-frequency IRE (H-FIRE) by Sano et al. [67]. Their in vitro study demonstrated that this combination results in lower overall cell viability compared to either therapy alone or the combination of unipolar IRE and RT. These findings highlight the potential of bipolar nsPEF-based ECT to enhance radiosensitization and prompt further investigation into its combination with RT. Therefore, more extensive research on this topic, especially in terms of bipolar sub-microsecond ECT, is needed.

To summarize, BPC affects the permeabilization of cells when 300/500 ns symmetrical pulses are used; however, the effects of pulse repetition frequency in the 10–1000 Hz range are negligible. We have also shown that the interphase delay of 0.1 ms is sufficient to diminish the effects of BPC on electroporation, and bipolar pulses can be used for CDDP-based ECT, similar to ESOPE bursts. Nevertheless, the in vitro and in vivo experiments should be matched up [68] since the BPC mechanisms described in vitro will not necessarily work the same way in vivo. A limitation of our study is that it does not evaluate the impact of shorter interphase delays (below 0.1 ms) as well as the impact of very long interphase delays. Specifically, when the negative phase approaches the positive phase of the next period, there is a higher possibility of BPC.

## 4. Materials and Methods

### 4.1. Cells

Murine hepatoma cells MH22a obtained from the Institute of Cytology of the Russian Academy of Sciences (St. Petersburg, Russia) were grown and maintained at 37 °C in 5% CO_2_ in DMEM medium, supplemented with 10% fetal bovine serum, 100 U/mL penicillin, and 0.1 mg/mL streptomycin, as described in previous work [69]. The experimental day MH22a cells were detached using Trypsin-EDTA solution, afterward centrifuged and resuspended in the DMEM medium, and further processed depending on the experiment. For the permeabilization and metabolic viability assays, cells were resuspended in the electroporation buffer (10 mM HEPES, 250 mM sucrose, and 1 mM MgCl_2_) at a concentration of 2 × 10^6^/mL, and for cytotoxicity determination in DMEM medium at a concentration of 5 × 10^4^/mL. All cell culture reagents were obtained from Gibco (Thermo Fisher Scientific, Grand Island, NY, USA).

### 4.2. Cell Permeabilization Detection Assay Using Fluorescent Markers

Electroporation-triggered murine hepatoma cells MH22a cell permeabilization was identified using the green-fluorescent dye Yo-Pro1 (YP, Sigma-Aldrich, St. Louis, MO, USA). Cells suspended in the electroporation buffer were combined with YP stain to achieve a final concentration of 1 μM. The 50 μL of mixed solution were placed between the electrodes and treated with various PEF protocols. Subsequently, the cells were transferred into a 96-well round bottom plate (Nunc, Sigma-Aldrich, St. Louis, MO, USA). Following a 3 min incubation at room temperature, 150 μL of 0.9% NaCl solution was added, and the samples were measured using a BD Accuri C6 flow cytometer (BD Biosciences, San Jose, CA, USA), where YP (Ex. 491⁄509) fluorescence was detected in Channel FL1 (Em. 533/30 nm BPF).

### 4.3. Viability Assay

To evaluate cisplatin (Accord, Dublin, Ireland) electrochemotherapy or the PEF-only effect, the viability of cells 24 h post-treatment was characterized using the PrestoBlue metabolic activity assay (Thermo Fisher Scientific, Waltham, MA, USA). A prepared MH22 cell suspension with cisplatin (25 μg/mL) was mixed and placed between the electrodes to be treated with different PEF conditions. Afterward, cells were transferred to a 96-well flat bottom plate (TPP, Trasadingen, Switzerland), where, after 10 min of incubation, 150 μL of growth media was added to each well, and the plate with samples was incubated for 24 h. The next day, the wells were washed twice with phosphate-buffered saline (PBS). Subsequently, 90 μL of PBS and 10 μL of cell viability reagent were added to each well, followed by a 2 h at 37 °C incubation period in the incubator. The fluorescence (Ex. 540/20 nm; Em. 620/40 nm) was measured using a Synergy 2 microplate reader and Gen5 software version 1.00.14 (BioTek, Shoreline, WA, USA).

### 4.4. Electroporation Setup and Parameters

The experimental configuration included a high-voltage pulse generator producing biphasic pulses ranging from 65 nanoseconds to 100 microseconds at voltages up to 3 kV, capable of generating bursts of pulses at a pre-defined frequency within the range of 1 Hz to 5 MHz [70]. This setup utilized a 1 mm gap commercially available electroporation cuvette (Biorad, Hercules, CA, USA). Firstly, in the permeabilization assay, fifty unipolar pulses with durations of 300 and 500 ns were delivered at repetition frequencies of 10 Hz, 100 Hz, and 1 kHz. Electric fields in the range of 4–13 kV/cm were used. The ESOPE protocol (1.2 kV/cm × 100 µs × 8, 1 Hz) was used to ensure comparability of the results of the whole study with established knowledge in the field. PEF protocols employed for the study are shown in Figure 6.

Subsequently, bipolar pulse bursts were employed in the study. Each bipolar pulse burst comprised a positive phase, an interphase delay, and a negative phase. Fifty pulses, composed of ↑300 ns + Δt + ↓300 ns and ↑500 ns + Δt + ↓500 ns with a variation of Δt from 0 to 95 ms, were used. We varied the interphase delay Δt between pulses of opposite polarities, selecting interphase delay values of 0, 0.1, 0.5, 1, 10, 20, 50, and 95 ms. The time between negative and positive phases was chosen based on the selected interphase delay value to maintain a sequence period of 0.1 s. For all PEF protocols, the total energized time was 5 s since a commonly used 10 Hz repetition frequency was employed. Figure 1 illustrates the representation of pulses utilized in this study. Further, the effects of pulse repetition frequency (10 Hz, 100 Hz, and 1 kHz with Period T of 100, 10, and 1 ms, respectively) were characterized with two 7 and 10 kV/cm PEF strengths. In these protocols, only 0, 0.1, and 0.5 ms of interphase delay were utilized since the pulse repetition frequency is higher (Figure 7A). Monophasic pulses of the same parameters were employed for efficacy comparison purposes (Figure 7B).

In order to determine the CDDP concentration to be used in ECT, cytotoxicity experiments (without PEF) were performed. The 5 × 10^4^/mL cells were seeded on microscope cover slides in plates either in the presence or absence of cisplatin (Accord, 1 mg/mL) of different concentrations. After 24 h of incubation, the slides were subsequently washed 3 times with phosphate-buffered saline and stained with Trypan blue. The cells adherent to the slides were counted under a light microscope. The viability of attached cells, determined by Trypan blue accumulation, was 98.5–99.3%. Cell viability after the compound treatment was expressed as the percentage of remaining adherent cells with respect to control. The concentration for 50% cell survival (lethal concentration; LC_50_) was calculated as shown in Figure 8.

It was found that a cisplatin concentration of 73.7 µg/mL (245 µM) is already an LC_50_ level (without external stimuli, cell viability reduces to 50%); therefore, for ECT studies, a lower concentration of 25 µg/mL (82 µM) was selected to ensure potentiation of the cytotoxicity by PEF-mediated intracellular delivery of CDDP.

### 4.5. Statistical Analysis

We utilized one-way analysis of variance (ANOVA; *p* < 0.05) to compare the findings. In cases where ANOVA revealed a statistically significant outcome (*p* < 0.05), the Tukey HSD multiple comparison test was employed to assess differences. Data were subsequently processed using OriginPro software version 9.8.0.200 (OriginLab, Northhampton, MA, USA). Each experiment was conducted a minimum of three times, and treatment efficiency was reported as mean ± standard deviation.

## 5. Conclusions

This study investigated the permeabilization of the MH-22a cell line and cisplatin-based electrochemotherapy using symmetrical bipolar pulsed electric fields with a varying interphase delay to overcome BCP. It was demonstrated that zero delay triggered the bipolar cancellation; however, there is a threshold interphase delay required to achieve YP uptake comparable to one triggered by unipolar PEF of the same parameters. Our results also showed that within the studied parametric range, the cancellation effect is not present in ECT, even though no interphase delay was introduced. The decreased cell membrane permeabilization (15–30%) due to BPC does not necessarily result in significant changes in ECT efficiency. Understanding the nuanced effects of different PEF parameters on permeabilization and viability is crucial for optimizing ECT protocols and improving treatment outcomes. These findings underscore the complex interplay between pulse parameters and their impact on cellular responses, paving the way for further refinement of ECT strategies for enhanced cancer therapy.

## Figures and Tables

**Figure 1 ijms-25-08774-f001:**
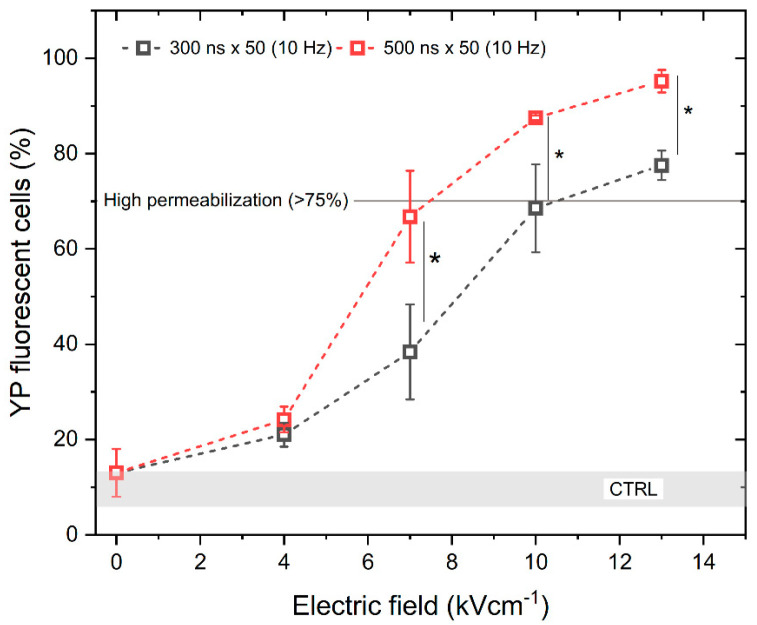
The relationship between cell membrane permeabilization, electric field intensity and pulse duration. All pulse sequences were composed of 50 pulses delivered at a 10 Hz pulse repetition frequency. The asterisk (*) corresponds to a statistically significant (*p* < 0.05) difference between samples. CTRL (shaded gray area) indicated the untreated control samples (9 ± 3% YP uptake).

**Figure 2 ijms-25-08774-f002:**
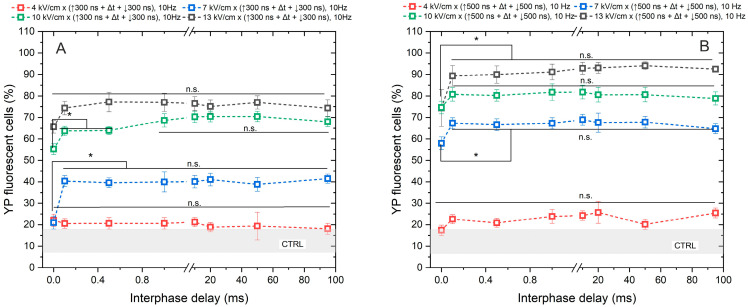
The dependence of cell membrane permeabilization on bipolar pulse parameters. Fifty electric field pulses were administered at a pulse repetition frequency of 10 Hz. Panels (**A**,**B**) depict the results for pulse durations of 300 ns and 500 ns, respectively. The asterisk (*) corresponds to a statistically significant (*p* < 0.05) difference between samples; n.s—statistically non-significant (*p* > 0.05). CTRL (shaded gray area) corresponds to the untreated control samples (12 ± 5% YP uptake).

**Figure 3 ijms-25-08774-f003:**
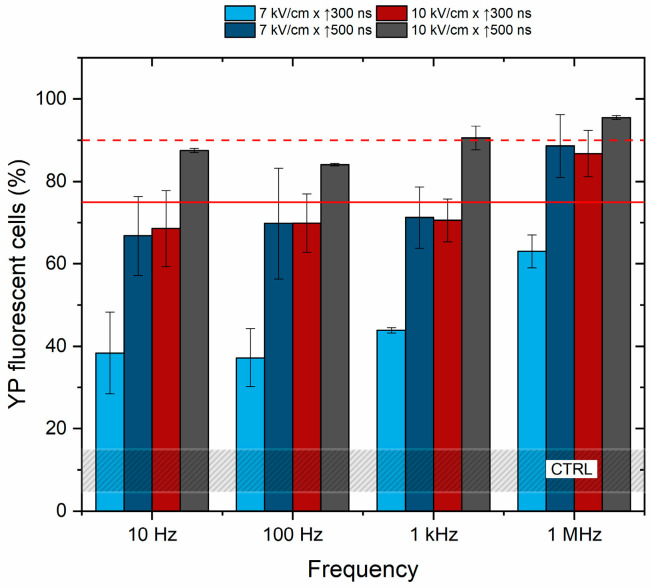
The dependence of cell membrane permeabilization on pulsed electric field parameters and pulse repetition frequency. The red dashed line shows permeabilization acquired by the reference ESOPE protocol (1.2 kV/cm × 100 μs × 8). The red solid line indicates a 75% permeabilization threshold, emphasizing the parameters that induce significant permeabilization, which are relevant for electrochemotherapy. CTRL (shaded gray area) corresponds to the untreated control samples (10 ± 5% YP uptake).

**Figure 4 ijms-25-08774-f004:**
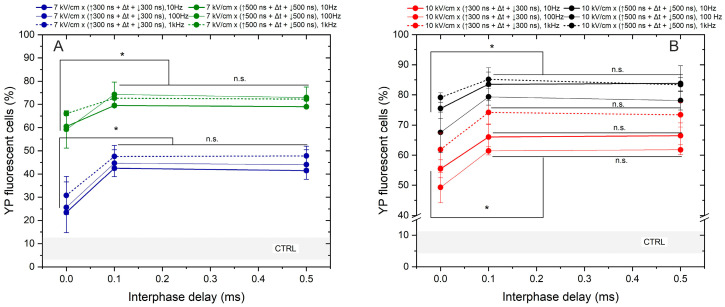
The dependence of cell membrane permeabilization on bipolar pulse parameters and pulse repetition frequency, where (**A**) 7 kV/cm electric field strength and (**B**) 10 kV/cm electric field strength. CTRL (shaded gray area) indicated the untreated control samples (8 ± 5% YP uptake). The asterisk (*) corresponds to a statistically significant (*p* < 0.05) difference between samples; n.s—statistically non-significant (*p* > 0.05).

**Figure 5 ijms-25-08774-f005:**
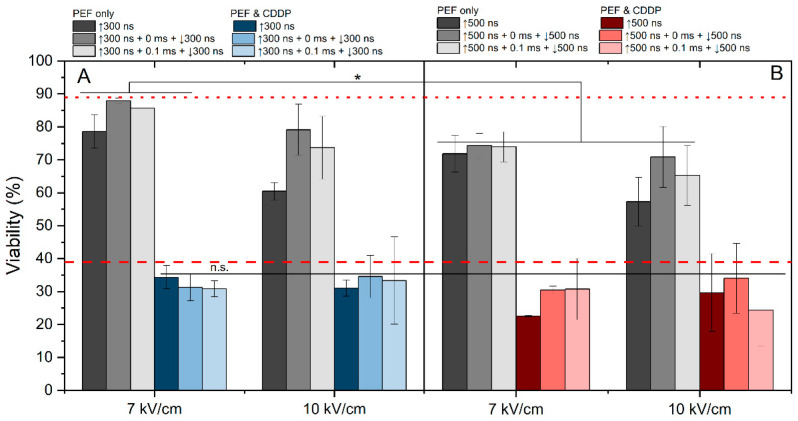
The dependence of ECT efficiency on applied PEF parameters after 24 h post-treatment, where (**A**) 7 kV/cm and 10 kV/cm efficiency with 300 ns pulse duration and (**B**) 7 V/cm and 10 kV/cm efficiency with 500 ns pulse duration. All pulse sequences were composed of 50 pulses delivered at a 10 Hz pulse repetition frequency. The red dotted line represents CDDP efficacy without PEF. The red dashed line represents average electrochemotherapy efficiency using the ESOPE protocol (1.2 kV/cm × 100 μs × 8). All data are normalized versus the untreated control (100 ± 10%). The asterisk (*) corresponds to a statistically significant (*p* < 0.05) difference between samples.

**Figure 6 ijms-25-08774-f006:**
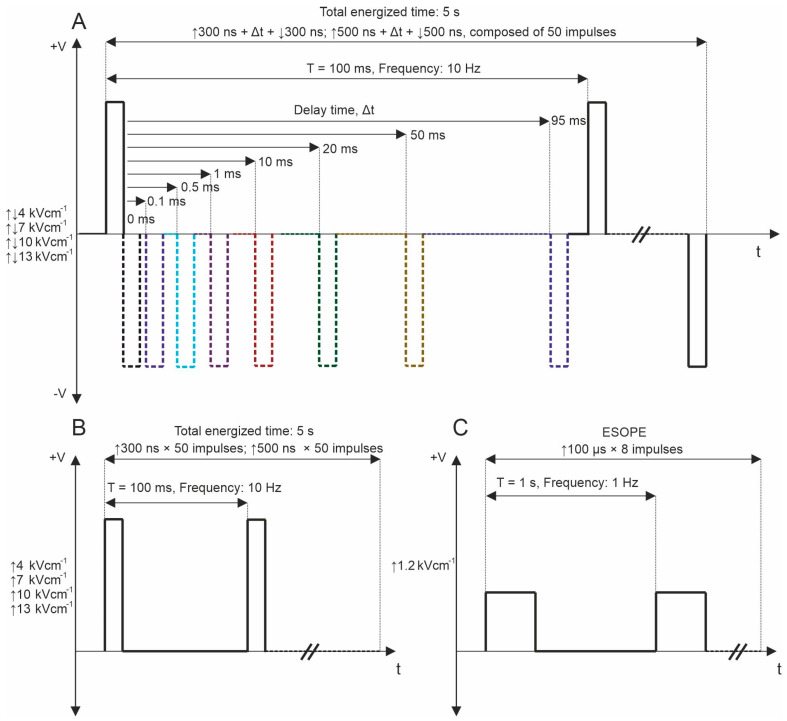
PEF protocols used for YP permeabilization, where (**A**) bipolar asymmetrical sequences; 50 pulses, which are composed of ↑300 ns + Δt + ↓300 ns and ↑500 ns + Δt + ↓500 ns with variation of Δt from 0 to 95 ms, delivered at 10 Hz repetition frequency; (**B**) unipolar sequences; 50 pulses, which are composed of ↑300 and ↑500, delivered at 10 Hz and 1 MHz repetition frequency; and (**C**) ESOPE (1.2 kV/cm × 100 µs × 8, 1 Hz). T refers to the period of the pulse sequence.

**Figure 7 ijms-25-08774-f007:**
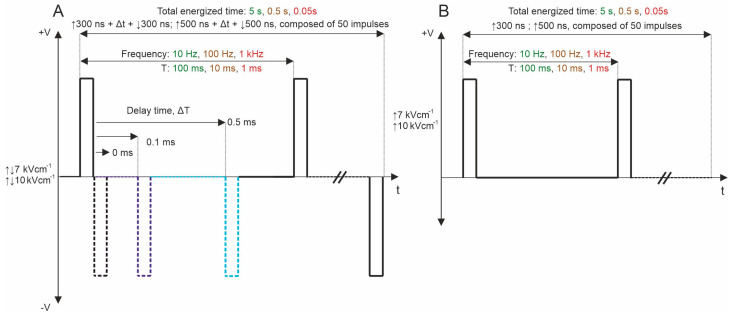
PEF protocols employed in the study: (**A**) biphasic asymmetrical pulse burst; (**B**) monophasic pulse burst. T refers to the period of the pulse sequence.

**Figure 8 ijms-25-08774-f008:**
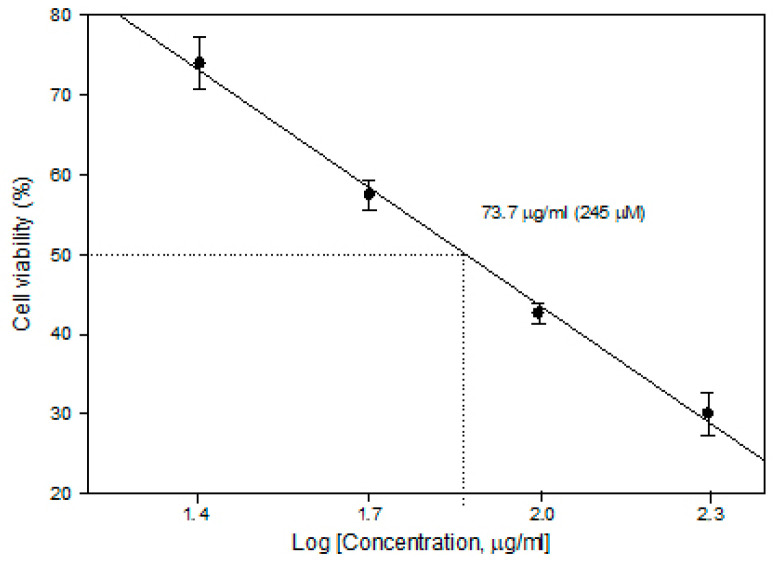
Cytotoxicity of cisplatin in the MH22a cell line.

## Data Availability

Data available from the corresponding author V.M-P. on request.

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
