# Peer review of "Threshold Interphase Delay for Bipolar Pulses to Prevent Cancellation Phenomenon during Electrochemotherapy"

_ijms, 2024, doi:10.3390/ijms25168774_

Round 1

Reviewer 1 Report

Comments and Suggestions for Authors

The authors provide a paper about “Threshold Interphase Delay for Bipolar Pulses to Prevent Cancellation Phenomenon During Electrochemotherapy”.

The topic is interesting both from a clinical and a research point of view.

I have a few points that I would like the authors to address as follows:

11)  Since the there are several parameters which could be manipulated from a technical point of view but with a strictly related biological (and clinical relevance) the authors in the introduction should more clearly and concisely state which are the biological and clinical advantages expected by the use of this technical manipulation (correlating it with biological considerations in terms of membrane permeability)

22) In the discussion section it would be an addition to add a paragraph dealing with the emerging evidence about the combination of Electrochemotherapy with Radiotherapy focusing on the topic of efficacy and potential radiosensitizing effects in tumor control

Author Response

Comments 1: The authors provide a paper about “Threshold Interphase Delay for Bipolar Pulses to Prevent Cancellation Phenomenon During Electrochemotherapy”.

The topic is interesting both from a clinical and a research point of view.

I have a few points that I would like the authors to address as follows:

11)  Since the there are several parameters which could be manipulated from a technical point of view but with a strictly related biological (and clinical relevance) the authors in the introduction should more clearly and concisely state which are the biological and clinical advantages expected by the use of this technical manipulation (correlating it with biological considerations in terms of membrane permeability)

Response 1: Thank you for pointing this out. We have clarified the relevance of our proposed technique on biological materials and clinical application.

Comments 2: In the discussion section it would be an addition to add a paragraph dealing with the emerging evidence about the combination of Electrochemotherapy with Radiotherapy focusing on the topic of efficacy and potential radiosensitizing effects in tumor control

Response 2: We have, accordingly, added a paragraph on this topic.

Reviewer 2 Report

Comments and Suggestions for Authors In this work, the authors investigate the influence of the interphase delay of electrical pulses on the permeabilisation of mouse hepatoma cells MH-22a during the application of bipolar nanosecond electrochemotherapy. They show that delays of more than 0.1 ms between positive and negative pulses are sufficient to minimise so-called bipolar cancellation phenomena. They also show that the efficiency of symmetrical bipolar CDDP-based electrochemotherapy is comparable to that of ESOPE procedures. My concerns are as follows: 1. Although the article is clearly written, some parts are difficult to follow. This is particularly true of Results chapter, which needs to be reorganised. For example, the description of Figure 1 is followed by a consideration of Figure 4. 2. It would greatly help the reader if the content of the research was briefly described at the end of the introduction. 3. The expected result for untreated cells (CTRL) is marked with a grey bar in the figures, but it is not described how these values were determined. Did the authors measure the permeabilisation of the untreated cells? 4. One of the main conclusions from this work is that there is a value for the interphase signal delay above which the bipolar cancellation effect is minimal. This conclusion is based on the measurement of permeabilisation for only two delay values, namely at 0 ms and 0.1 ms. This also means that the data from the delay interval between 0 and 0.1 ms are the most relevant to support such a conclusion. It is not clear why the authors did not measure at least one other relevant value, for example for a delay of 0.05 ms, but a range of values for longer delays where there is no permeabilisation change at all. 5. The authors claim that the dependence of cell permeabilisation on pulse frequency is not significant. However, for all four combinations of electric field strength and pulse width shown in Figure 4, the same frequency dependence of cell permeabilisation can be seen. Permeabilisation is always highest at a pulse frequency of 1 kHz and lowest at 100 Hz. This does not seem to be a coincidence and is worth considering. Moreover, the frequency dependence in the case of 10 kV and 300 ns is comparable to the bipolar cancellation effect, i.e. the change in delay from zero to 0.1 ms. 6. The data and markings in Figures 6 and 7 should be corrected or explained. The time intervals between pulses should be indicated by the duration in milliseconds, not by the frequency in hertz. The formula for the energization time 300 ns + 0.1 ms + 300 ns × 50 pulses is misleading. I suggest to write e.g. 50 pulses, which are composed of 300 ns + ΔT + 300 ns intervals. Please correct this for other images as well. Also, the time intervals are usually labelled Δt, while ΔT is the label for the temperature difference. 7. All abbreviations must be defined the first time they occur. Comments on the Quality of English Language

Author Response

In this work, the authors investigate the influence of the interphase delay of electrical pulses on the permeabilisation of mouse hepatoma cells MH-22a during the application of bipolar nanosecond electrochemotherapy. They show that delays of more than 0.1 ms between positive and negative pulses are sufficient to minimise so-called bipolar cancellation phenomena. They also show that the efficiency of symmetrical bipolar CDDP-based electrochemotherapy is comparable to that of ESOPE procedures. My concerns are as follows:

Comments 1: Although the article is clearly written, some parts are difficult to follow. This is particularly true of Results chapter, which needs to be reorganised. For example, the description of Figure 1 is followed by a consideration of Figure 4.

Response 1: Thank you for pointing this out. We agree with this comment. This was due to typing mistake. Therefore, we have edited the mentioned text.

Comments 2: It would greatly help the reader if the content of the research was briefly described at the end of the introduction.

Response 1: Thank you for pointing this out. However, the last paragraph of Introduction chapter has covered the main content of the research.

“In this study, for the first time, we characterized the effect of interphase durations ranging from 0 to 95 ms with a 10 Hz pulse repetition frequency, and from 0 to 0.5 ms with 100 Hz and 1 kHz bursts, respectively. Our results showed that there is a threshold delay value that allows mitigation of the bipolar cancellation effect in all the tested frequency ranges, enabling to achieve molecular uptake equivalent to that obtained with unipolar pulses. Subsequently, CDDP-based ECT efficiency employing symmetrical bipolar and unipolar pulses was compared to ESOPE procedures.”

Thought, we have clarified this paragraph and highlighted the text in red.

Comments 3: The expected result for untreated cells (CTRL) is marked with a grey bar in the figures, but it is not described how these values were determined. Did the authors measure the permeabilisation of the untreated cells?

Response 1: Thank you for pointing this out. Yes, the permeabilization of untreated cells was measured for each experiment. Consequently, the value range within untreated samples is depicted as a grey shaded area, representing STDEV of acquired permeabilization in CTRL samples from independent experiments incl. repetitions.

Comments 4: One of the main conclusions from this work is that there is a value for the interphase signal delay above which the bipolar cancellation effect is minimal. This conclusion is based on the measurement of permeabilisation for only two delay values, namely at 0 ms and 0.1 ms. This also means that the data from the delay interval between 0 and 0.1 ms are the most relevant to support such a conclusion. It is not clear why the authors did not measure at least one other relevant value, for example for a delay of 0.05 ms, but a range of values for longer delays where there is no permeabilisation change at all.

Response 1: We have relied on established knowledge and published research. There are papers indicating that 0.05 – 0.15 ms delays are still causing cancellation (cited in our work). Using protocols, which already trigger cancellation has no clinical relevance for ECT. Therefore, we have used 0 delay, to highlight the presence of the phenomenon and to provide motivation/relevance for the whole study. Then we planned an overlapping point 0.1 ms with other studies, to ensure consolidation of knowledge and comparison and further increased the delay to the ranges, which were not yet covered in literature. When we got experimental proof that indeed these delays are “safe”, for the first time we characterized the effects of pulse repetition frequency on the phenomenon. Also, there is hardly any research of cancellation in the context of ECT. To summarize, our study aimed for applied aspects of ECT to further move relevant protocols to in vivo, rather than research of the range, where there is already evidence that cancellation can occur and thus these protocols are sub-optimal.

Comments 5: The authors claim that the dependence of cell permeabilisation on pulse frequency is not significant. However, for all four combinations of electric field strength and pulse width shown in Figure 4, the same frequency dependence of cell permeabilisation can be seen. Permeabilisation is always highest at a pulse frequency of 1 kHz and lowest at 100 Hz. This does not seem to be a coincidence and is worth considering. Moreover, the frequency dependence in the case of 10 kV and 300 ns is comparable to the bipolar cancellation effect, i.e. the change in delay from zero to 0.1 ms.

Response 1: Indeed, there is a tendency that the average value of permeabilization can depend on frequency, however, in many cases the differences are not statistically significant. In cases, when it is, the difference is usually in ~10% range, which is significantly lower when compared to the cancellation effects. Considering that for majority of the involved protocols the frequency is a minor factor, we have performed ECT using 10 Hz frequency (very commonly used in studies involving bursts of ns pulses) to ensure repeatability by other groups. However, to account for the comment we have extended our discussion to highlight that it’s indicatory that the effects of frequency can be also exploited in the future for optimization of the protocols.

Comments 6: The data and markings in Figures 6 and 7 should be corrected or explained. The time intervals between pulses should be indicated by the duration in milliseconds, not by the frequency in hertz. The formula for the energization time 300 ns + 0.1 ms + 300 ns × 50 pulses is misleading. I suggest to write e.g., 50 pulses, which are composed of 300 ns + ΔT + 300 ns intervals. Please correct this for other images as well. Also, the time intervals are usually labelled Δt, while ΔT is the label for the temperature difference.

Response 1: Thank you for pointing this out. We agree with this comment. Therefore, we have updated the Figures, its captions and description in manuscript body.

Comments 7: All abbreviations must be defined the first time they occur.

Response 1: Thank you for pointing this out. We agree with this comment. Therefore, we have corrected the abbreviations.

Round 2

Reviewer 2 Report

Comments and Suggestions for Authors

Comments 1: Although the article is clearly written, some parts are difficult to follow. This is particularly true of Results chapter, which needs to be reorganised. For example, the description of Figure 1 is followed by a consideration of Figure 4.

Response 1: Thank you for pointing this out. We agree with this comment. This was due to typing mistake. Therefore, we have edited the mentioned text.

Comment 1a: The text is now even more difficult to read because the figures are no longer labelled with serial numbers.

Comments 2: It would greatly help the reader if the content of the research was briefly described at the end of the introduction.

Response 1: Thank you for pointing this out. However, the last paragraph of Introduction chapter has covered the main content of the research.

“In this study, for the first time, we characterized the effect of interphase durations ranging from 0 to 95 ms with a 10 Hz pulse repetition frequency, and from 0 to 0.5 ms with 100 Hz and 1 kHz bursts, respectively. Our results showed that there is a threshold delay value that allows mitigation of the bipolar cancellation effect in all the tested frequency ranges, enabling to achieve molecular uptake equivalent to that obtained with unipolar pulses. Subsequently, CDDP-based ECT efficiency employing symmetrical bipolar and unipolar pulses was compared to ESOPE procedures.”

Thought, we have clarified this paragraph and highlighted the text in red.

Comment 2a:

In the abstract, the authors claim: “Therefore, in this work we have tested the influence of different interphase delay values (from 0 ms to 95 ms) using symmetric bipolar nanosecond (300- and 500 ns) electrochemotherapy (in vitro) with cisplatin using 10 Hz, 100 Hz and 1 kHz protocols.” At the beginning of the discussion, the authors state: “This study is focused on investigating the role of different interphase delay values on BPC and efficiency of cisplatin-based ECT in vitro using mouse hepatoma MH-22a cellline as a model.” However, the conclusion state: „This study investigated cisplatin-based electrochemotherapy using symmetrical bipolar pulsed electric fields with a varying interphase delay.“ It is certainly necessary for authors to express correctly and clearly what they did and how they did it, because the main part of the work was to investigate the effect of delay on permeabilisation using fluorescent markers.

Comments 3: The expected result for untreated cells (CTRL) is marked with a grey bar in the figures, but it is not described how these values were determined. Did the authors measure the permeabilisation of the untreated cells?

Response 1: Thank you for pointing this out. Yes, the permeabilization of untreated cells was measured for each experiment. Consequently, the value range within untreated samples is depicted as a grey shaded area, representing STDEV of acquired permeabilization in CTRL samples from independent experiments incl. repetitions.

Comments 3a: The control sample data are very important to determine the measurement error. Nothing is said about this in the manuscript. It is necessary to describe and to show the control sample data for all images and to give numerical values for the reliability of the data.

Comments 4: One of the main conclusions from this work is that there is a value for the interphase signal delay above which the bipolar cancellation effect is minimal. This conclusion is based on the measurement of permeabilisation for only two delay values, namely at 0 ms and 0.1 ms. This also means that the data from the delay interval between 0 and 0.1 ms are the most relevant to support such a conclusion. It is not clear why the authors did not measure at least one other relevant value, for example for a delay of 0.05 ms, but a range of values for longer delays where there is no permeabilisation change at all.

Response 1: We have relied on established knowledge and published research. There are papers indicating that 0.05 – 0.15 ms delays are still causing cancellation (cited in our work). Using protocols, which already trigger cancellation has no clinical relevance for ECT. Therefore, we have used 0 delay, to highlight the presence of the phenomenon and to provide motivation/relevance for the whole study. Then we planned an overlapping point 0.1 ms with other studies, to ensure consolidation of knowledge and comparison and further increased the delay to the ranges, which were not yet covered in literature. When we got experimental proof that indeed these delays are “safe”, for the first time we characterized the effects of pulse repetition frequency on the phenomenon. Also, there is hardly any research of cancellation in the context of ECT. To summarize, our study aimed for applied aspects of ECT to further move relevant protocols to in vivo, rather than research of the range, where there is already evidence that cancellation can occur and thus these protocols are sub-optimal.

Comment 4a: In line 216, the authors claim: “We have shown that a cancellation effect is present when the interphase delay is low, however, once the delay exceeds 0.1 ms, the efficacy of bipolar pulses becomes comparable with unipolar ones.” is not evident. Only two values were measured, 0 ms and 0.1 ms, and therefore it is not known at which delay the cancellation effect occurs. Furthermore, it is unreliable to draw such a conclusion from these two values alone. The authors should numerically compare the magnitude of the effect obtained with the experimental error and clearly state the limitations of this experiment and possible conclusions.

Comments 5: The authors claim that the dependence of cell permeabilisation on pulse frequency is not significant. However, for all four combinations of electric field strength and pulse width shown in Figure 4, the same frequency dependence of cell permeabilisation can be seen. Permeabilisation is always highest at a pulse frequency of 1 kHz and lowest at 100 Hz. This does not seem to be a coincidence and is worth considering. Moreover, the frequency dependence in the case of 10 kV and 300 ns is comparable to the bipolar cancellation effect, i.e. the change in delay from zero to 0.1 ms.

Response 1: Indeed, there is a tendency that the average value of permeabilization can depend on frequency, however, in many cases the differences are not statistically significant. In cases, when it is, the difference is usually in ~10% range, which is significantly lower when compared to the cancellation effects. Considering that for majority of the involved protocols the frequency is a minor factor, we have performed ECT using 10 Hz frequency (very commonly used in studies involving bursts of ns pulses) to ensure repeatability by other groups. However, to account for the comment we have extended our discussion to highlight that it’s indicatory that the effects of frequency can be also exploited in the future for optimization of the protocols.

Comment 5a. The authors caims: “The differences in permeabilization efficacy between 10, 100 Hz and 1 kHz pulses is usually in sub–10% range” in line 185.and “the results from our permeabilization assays indicate that frequency is a minor factor within the tested pulsing protocols” in line 192, are not confirmed by the results shown in Figure 4B. In order to draw such conclusions, the considered effects should be numerically compared and contrasted with the experimental error.

Comments 6: The data and markings in Figures 6 and 7 should be corrected or explained. The time intervals between pulses should be indicated by the duration in milliseconds, not by the frequency in hertz. The formula for the energization time 300 ns + 0.1 ms + 300 ns × 50 pulses is misleading. I suggest to write e.g., 50 pulses, which are composed of 300 ns + ΔT + 300 ns intervals. Please correct this for other images as well. Also, the time intervals are usually labelled Δt, while ΔT is the label for the temperature difference.

Response 1: Thank you for pointing this out. We agree with this comment. Therefore, we have updated the Figures, its captions and description in manuscript body.

Comments 6a: As for the images, there are still a few things that need to be fixed to make it easier to read and draw conclusions. Statistically significant differences between samples are shown graphically in all figures, and it is left to the reader to judge the significance of the treatment effect. However, for several images it is not clear whether the effect of the treatment is large enough in relation to the error to draw conclusions about the effect. Therefore, in addition to the graphical values, it is necessary to provide numerical values that make it much clearer. This applies in particular to Figure 2 and Figure 4.

There are still a large number of formulae in the text and in the pictures that are not mathematically correct. For example the data labels in Figure 4 are given in the form of expressions in which the multiplication sign (x) is used incorrectly.

In addition;

Lines 134, 356, 358, 362 The time unit is missing in the formula

Line 205 (1.2 kV/cm x 205 100 μs x 8)

Line 359: The meaning of the expression (1.2 358 kV/cm × 100 μs × 8.1 Hz) is unclear.

Comments on the Quality of English Language

Author Response

Thank you very much for taking the time to review this manuscript. Please find the detailed responses below and the corresponding revisions/corrections highlighted in blue.

Comments 1: Although the article is clearly written, some parts are difficult to follow. This is particularly true of Results chapter, which needs to be reorganised. For example, the description of Figure 1 is followed by a consideration of Figure 4.

Response 1: Thank you for pointing this out. We agree with this comment. This was due to typing mistake. Therefore, we have edited the mentioned text.

Comment 1a: The text is now even more difficult to read because the figures are no longer labelled with serial numbers.

Response 1a: We have checked the uploaded manuscript, all labels were provided, however, when downloaded the latest version of manuscript they were all gone probably due to conversion from docx to PDF. We have rechecked and updated all labels in latest manuscript.

Comments 2: It would greatly help the reader if the content of the research was briefly described at the end of the introduction.

Response 2: Thank you for pointing this out. However, the last paragraph of Introduction chapter has covered the main content of the research.

“In this study, for the first time, we characterized the effect of interphase durations ranging from 0 to 95 ms with a 10 Hz pulse repetition frequency, and from 0 to 0.5 ms with 100 Hz and 1 kHz bursts, respectively. Our results showed that there is a threshold delay value that allows mitigation of the bipolar cancellation effect in all the tested frequency ranges, enabling to achieve molecular uptake equivalent to that obtained with unipolar pulses. Subsequently, CDDP-based ECT efficiency employing symmetrical bipolar and unipolar pulses was compared to ESOPE procedures.” We have clarified and improved this paragraph.

Comment 2a:

In the abstract, the authors claim: “Therefore, in this work we have tested the influence of different interphase delay values (from 0 ms to 95 ms) using symmetric bipolar nanosecond (300- and 500 ns) electrochemotherapy (in vitro) with cisplatin using 10 Hz, 100 Hz and 1 kHz protocols.” At the beginning of the discussion, the authors state: “This study is focused on investigating the role of different interphase delay values on BPC and efficiency of cisplatin-based ECT in vitro using mouse hepatoma MH-22a cellline as a model.” However, the conclusion state: „This study investigated cisplatin-based electrochemotherapy using symmetrical bipolar pulsed electric fields with a varying interphase delay.“ It is certainly necessary for authors to express correctly and clearly what they did and how they did it, because the main part of the work was to investigate the effect of delay on permeabilisation using fluorescent markers.

Response 2a: Permeabilization and viability are the two fundamental parameters, which must be characterized in any applied electroporation-based study focusing molecular delivery. The protocols, which trigger saturated permeabilization (90%+) and reversible electroporation (minor effects on cell viability) are the most suitable for electrochemotherapy. In case of bipolar pulses, the cancellation phenomenon requires compensation/solution for the burst to be applicable in ECT context, which in our case is analyzed from the perspective of delay between consequent opposite phases of the pulses. Therefore, firstly we characterize protocols based on permeabilization, which is the main parameter of electroporation and later the best protocols are tested for ECT where the viability is evaluated. Permeabilization and viability are inseparable when it comes to electroporation. Also, permeabilization is a pre-requisite for effective molecular delivery. Therefore, lots of attention in the paper is dedicated for permeabilization using fluorescent markers, which is in absolute agreement with the standards and recommendations for reporting electroporation research.

Comments 3: The expected result for untreated cells (CTRL) is marked with a grey bar in the figures, but it is not described how these values were determined. Did the authors measure the permeabilisation of the untreated cells?

Response 3: Thank you for pointing this out. Yes, the permeabilization of untreated cells was measured for each experiment. Consequently, the value range within untreated samples is depicted as a grey shaded area, representing STDEV of acquired permeabilization in CTRL samples from independent experiments incl. repetitions.

Comments 3a: The control sample data are very important to determine the measurement error. Nothing is said about this in the manuscript. It is necessary to describe and to show the control sample data for all images and to give numerical values for the reliability of the data.

Response 3a: We have added numerical value of CTRL with STDEV to each Figure caption.

Comments 4: One of the main conclusions from this work is that there is a value for the interphase signal delay above which the bipolar cancellation effect is minimal. This conclusion is based on the measurement of permeabilisation for only two delay values, namely at 0 ms and 0.1 ms. This also means that the data from the delay interval between 0 and 0.1 ms are the most relevant to support such a conclusion. It is not clear why the authors did not measure at least one other relevant value, for example for a delay of 0.05 ms, but a range of values for longer delays where there is no permeabilisation change at all.

Response 4: We have relied on established knowledge and published research. There are papers indicating that 0.05 – 0.15 ms delays are still causing cancellation (cited in our work). Using protocols, which already trigger cancellation has no clinical relevance for ECT. Therefore, we have used 0 delay, to highlight the presence of the phenomenon and to provide motivation/relevance for the whole study. Then we planned an overlapping point 0.1 ms with other studies, to ensure consolidation of knowledge and comparison and further increased the delay to the ranges, which were not yet covered in literature. When we got experimental proof that indeed these delays are “safe”, for the first time we characterized the effects of pulse repetition frequency on the phenomenon. Also, there is hardly any research of cancellation in the context of ECT. To summarize, our study aimed for applied aspects of ECT to further move relevant protocols to in vivo, rather than research of the range, where there is already evidence that cancellation can occur and thus these protocols are sub-optimal.

Comment 4a: In line 216, the authors claim: “We have shown that a cancellation effect is present when the interphase delay is low, however, once the delay exceeds 0.1 ms, the efficacy of bipolar pulses becomes comparable with unipolar ones.” is not evident. Only two values were measured, 0 ms and 0.1 ms, and therefore it is not known at which delay the cancellation effect occurs. Furthermore, it is unreliable to draw such a conclusion from these two values alone. The authors should numerically compare the magnitude of the effect obtained with the experimental error and clearly state the limitations of this experiment and possible conclusions.

Response 4a: The evidence and reliability of data is characterized by statistical tests (in this case ANOVA + post-hoc Tukey HSD). If the difference between two parameters is statistically significant (P<0.05) it means the result is reliably different. We marked reliable differences in each graph characterizing the diminishing of cancellation effect when the delay increases. Subsequently the overview of the graph in the text and strong specific statements are formed based on the statistical data, which is a standard scientific practice and the proof of reliability. Also, the maximum cancellation (established knowledge) happens when the delay between phases is 0. In our case, we have this marginal case and the cancellation is 10-30% in terms of permeabilization. The typical STDEV for independent biological repetitions of permeabilization experiments is often ±5-10%. Therefore, having smaller delay steps is just non-relevant, since the significance of the result will be lost in the STDEV of data. Finally, we empirically show that sub-20% improvements in permeabilization are in-consequential for applied research (i.e., ECT), therefore, the scheme was developed as it is – test ranges, which were not yet covered in literature focusing the applied potential of the different bursts (incl. delay, amplitude, frequency modulation) rather than protocols, which are already expected to be sub-optimal due to cancellation.

Comments 5: The authors claim that the dependence of cell permeabilisation on pulse frequency is not significant. However, for all four combinations of electric field strength and pulse width shown in Figure 4, the same frequency dependence of cell permeabilisation can be seen. Permeabilisation is always highest at a pulse frequency of 1 kHz and lowest at 100 Hz. This does not seem to be a coincidence and is worth considering. Moreover, the frequency dependence in the case of 10 kV and 300 ns is comparable to the bipolar cancellation effect, i.e. the change in delay from zero to 0.1 ms.

Response 5: Indeed, there is a tendency that the average value of permeabilization can depend on frequency, however, in many cases the differences are not statistically significant. In cases, when it is, the difference is usually in ~10% range, which is significantly lower when compared to the cancellation effects. Considering that for majority of the involved protocols the frequency is a minor factor, we have performed ECT using 10 Hz frequency (very commonly used in studies involving bursts of ns pulses) to ensure repeatability by other groups. However, to account for the comment we have extended our discussion to highlight that it’s indicatory that the effects of frequency can be also exploited in the future for optimization of the protocols.

Comment 5a. The authors caims: “The differences in permeabilization efficacy between 10, 100 Hz and 1 kHz pulses is usually in sub–10% range” in line 185.and “the results from our permeabilization assays indicate that frequency is a minor factor within the tested pulsing protocols” in line 192, are not confirmed by the results shown in Figure 4B. In order to draw such conclusions, the considered effects should be numerically compared and contrasted with the experimental error.

Response 5a: We compare all of the results numerically. That is why statistical significance tests are performed. If the result is statistically significant it’s marked with an asterisk. In all graphs this analysis is presented. Basically, in case of frequency dependence there are some instances when frequency affects permeabilization. While the differences in some cases return statistical significance (P<0.05) the applied aspects of ~15% range permeabilization modulation are minor or even non-existent (see viability data). That is why we claim that “frequency is a minor factor within the tested pulsing protocols”.

Comments 6: The data and markings in Figures 6 and 7 should be corrected or explained. The time intervals between pulses should be indicated by the duration in milliseconds, not by the frequency in hertz. The formula for the energization time 300 ns + 0.1 ms + 300 ns × 50 pulses is misleading. I suggest to write e.g., 50 pulses, which are composed of 300 ns + ΔT + 300 ns intervals. Please correct this for other images as well. Also, the time intervals are usually labelled Δt, while ΔT is the label for the temperature difference.

Response 6: Thank you for pointing this out. We agree with this comment. Therefore, we have updated the Figures, its captions and description in manuscript body.

Comments 6a: As for the images, there are still a few things that need to be fixed to make it easier to read and draw conclusions. Statistically significant differences between samples are shown graphically in all figures, and it is left to the reader to judge the significance of the treatment effect. However, for several images it is not clear whether the effect of the treatment is large enough in relation to the error to draw conclusions about the effect. Therefore, in addition to the graphical values, it is necessary to provide numerical values that make it much clearer. This applies in particular to Figure 2 and Figure 4.

There are still a large number of formulae in the text and in the pictures that are not mathematically correct. For example the data labels in Figure 4 are given in the form of expressions in which the multiplication sign (x) is used incorrectly.

In addition;

Lines 134, 356, 358, 362 The time unit is missing in the formula

Line 205 (1.2 kV/cm x 205 100 μs x 8)

Line 359: The meaning of the expression (1.2 358 kV/cm × 100 μs × 8.1 Hz) is unclear.

Response 6a: We have edited the description of numerical data represented in Figures 2. Description of Figure 4 already covers the exact values form graph. The results shown in the figures indicate that 0 ms delay (or no delay) triggers cancellation effect, 0.1 ms and longer delay minimized BCP. For all tested protocols YP uptake values varies, thus adding all values to graphs would make it bulky and hardly readable.

Regarding the Figure 2 and Figure 4 , we agree that legend should include brackets and interphase delay Δt e.g., 7 kV/cm x (↑300 + Δt + ↓300 ), 10 Hz. As for expression for the ESOPE protocol, it is typically represented in a way given in this manuscript. Please, check the references of papers where expression is in agreement with ours.

  1. Scuderi, M., Dermol-Cerne, J., Scancar, J., Markovic, S., Rems, L., & Miklavcic, D. (2024). The equivalence of different types of electric pulses for electrochemotherapy with cisplatin− an study. Radiology and Oncology58(1), 51-66.
  2. Kulbacka, J., Rembiałkowska, N., Szewczyk, A., Moreira, H., Szyjka, A., Girkontaitė, I., ... & Novickij, V. (2021). The impact of extracellular Ca2+ and nanosecond electric pulses on sensitive and drug-resistant human breast and colon cancer cells. Cancers13(13), 3216.
  3. Dermol-Černe, J., Vidmar, J., Ščančar, J., Uršič, K., Serša, G., & Miklavčič, D. (2018). Connecting the in vitro and in vivo experiments in electrochemotherapy-a feasibility study modeling cisplatin transport in mouse melanoma using the dual-porosity model. Journal of controlled release286, 33-45.
  4. Frandsen, S. K., McNeil, A. K., Novak, I., McNeil, P. L., & Gehl, J. (2016). Difference in membrane repair capacity between cancer cell lines and a normal cell line. The Journal of Membrane Biology249, 569-576.

And many others.

Round 3

Reviewer 2 Report

Comments and Suggestions for Authors In their replies to Comment 2 the authors explain to me, quite unnecessarily, how and what they have done, instead of doing so in the text of the manuscript. Furthermore, in accordance with some of my comments, the authors have made certain changes in the text that they did not mention in the response to the comments.
In response to repeated requests, the authors do not provide data for the control samples, which in my opinion is a major problem for the scientific validity of the manuscript.
In their response to comment 4, the authors claim that smaller lag steps are simply not relevant because the significance of the result is lost in the STDEV of the data. This implies that their results are at the threshold of significance and only reinforces my concerns about the reliability of the results.
It remains unclear why the authors claim that the dependence of cell permeabilisation on pulse frequency is not significant when this dependence can be clearly seen in Figure 4. Comments on the Quality of English Language

Author Response

In their replies to Comment 2 the authors explain to me, quite unnecessarily, how and what they have done, instead of doing so in the text of the manuscript. Furthermore, in accordance with some of my comments, the authors have made certain changes in the text that they did not mention in the response to the comments. 

Reviewer: In response to repeated requests, the authors do not provide data for the control samples, which in my opinion is a major problem for the scientific validity of the manuscript. 

Comment: We provided all of the numerical data of untreated control samples in each graph caption during previous revision. E.g., “CTRL corresponds to the untreated control samples (8±5% YP uptake)”. Absolutely all the graphs have crucial information about control samples as was requested by the reviewer in previous revision round. Therefore, we have addressed the remark previously.

Reviewer: In their response to comment 4, the authors claim that smaller lag steps are simply not relevant because the significance of the result is lost in the STDEV of the data. This implies that their results are at the threshold of significance and only reinforces my concerns about the reliability of the results. 

Comment: Cancellation phenomenon within the range of tested parameters influences up to 30% of permeabilization (marginal case with 0 delay). We show experimental data that while such a change influences permeabilization (P<0.01), the impact on electrochemotherapy is non-existent (no statistically significant changes in cell viability). It’s an important result for applied aspects of ECT using bipolar pulses. Therefore, “playing” with lower delays, has no practical significance at all within the repetition frequency ranges tested. Also taking into account that the marginal case is 30% change of YP uptake, of course going to lower delays will induce smaller changes, which is within typical STDEV for reporting permeabilization. The repeatability by other groups will be hindered. Especially taking into account that the threshold will be influenced by the cell type, their condition, buffer composition. That is why some available works report that 50 us is enough, others that cancellation is still observable with 150 us delays. Our work is different, we answer a question, which is frequently forgotten: “But does it have an influence for real applications like ECT?” And the answer is no (at least in vitro), within the range of repetition frequencies covered. We show that when you reach a “safe” threshold (the efficiency of permeabilization is comparable to a unipolar pulse, in our case 100 us), the further increase of delay does not play a role on permeabilization and of course ECT. Additionally, even 0 delay (marginal case) is applicable for ECT when the repetition frequency is low. If we saw that cancellation phenomenon had a significant effect on ECT, of course we would dive deeper in terms of thresholds – but this is not the case and has not practical applicability.

Reviewer: It remains unclear why the authors claim that the dependence of cell permeabilisation on pulse frequency is not significant when this dependence can be clearly seen in Figure 4.

Comment: Indeed, one can see some difference in average value, but it’s not statistically significant, which we mark in the graph. If statistical tests do not return statistical significance of at least P<0.05 the result cannot be interpreted as “different”. Therefore, our claims are based on appropriate ANOVA and post-hoc Tukey HSD multiple comparison test.